# Role of Biofunctionalized Nanoparticles in Digestive Cancer Vaccine Development

**DOI:** 10.3390/pharmaceutics16030410

**Published:** 2024-03-16

**Authors:** Razvan Zdrehus, Cristian Delcea, Lucian Mocan

**Affiliations:** 1Doctoral School, Iuliu Hatieganu University of Medicine and Pharmacy, 400000 Cluj-Napoca, Romania; 2Department of Forensic Medicine, Iuliu Hatieganu University of Medicine and Pharmacy, 400000 Cluj-Napoca, Romania; 3Department of Surgery, 3-rd Surgery Clinic, Iuliu Hatieganu University of Medicine and Pharmacy, 400000 Cluj-Napoca, Romania; mocanlucian@yahoo.com

**Keywords:** cancer, vaccine, nanoparticle, immunotherapy

## Abstract

Nanotechnology has provided an opportunity for unparalleled development of the treatment of various severe diseases. The unique properties of nanoparticles offer a promising strategy for enhancing antitumor immunity by enhancing immunogenicity and presentation of tumor autoantigens for cancer immunotherapy. Polymeric, liposomal, carbon or silica-based nanoparticles are among those with major immunomodulatory roles in various cancer treatments. Cancer vaccines, in particular digestive cancer vaccines, have been researched and developed on nanotechnological platforms. Due to their safety, controlled release, targeting of dendritic cells (DCs) and improved antigen uptake, as well as enhanced immunogenicity, nanoparticles have been used as carriers, as adjuvants for increased effect at the tumor level, for their immunomodulating effect, or for targeting the tumor microenvironment, thereby increasing tumor immunogenicity and reducing tumor inflammatory response. This review looks at digestive cancer vaccines developed on nanoparticle platforms and the impact nanoparticles have on the effects of these vaccines.

## 1. Introduction

The use of nanotechnology in medicine has provided an opportunity for unparalleled development of the treatment of various severe diseases. The unique properties of nanoparticles (NPs), such as small size and higher surface-to-volume ratio, the ability to encapsulate different drugs and tunable surface chemistry, offer many advantages over already existing technologies: efficient navigation of the nanoparticle complex in vivo, multivalent surface modification for greater accuracy, increased intracellular trafficking, addition of charged particles to increase target selectivity, or sustained drug release. Nanoparticles can thus be an ideal candidate in medical applications for the most widespread and challenging health problems, such as cancer. Nanotechnology offers a promising strategy for enhancing antitumor immunity by enhancing immunogenicity and presentation of tumor autoantigens for cancer immunotherapy.

The term “cancer vaccine” denotes a vaccination strategy aimed at either averting infections caused by cancer-inducing viruses or impeding cancer development in predisposed individuals (referred to as prophylactic cancer vaccines) or, alternatively, addressing existing malignancies (referred to as therapeutic cancer vaccines) [1]. The integration of nanoparticles in the development of new vaccines presents significant advantages, facilitating more precise targeting of antigen-presenting cells [2,3]. Depending on the mechanism of action, anticancer vaccines can be categorized into three main groups: cellular vaccines, protein or peptide vaccines, and genetic vaccines encompassing RNA, DNA, or viral particles (see Figure 1) [4].

Cell vaccines can use autologous tumor cells, which are rendered inactive through irradiation and typically combined with adjuvant molecules. The mechanism of action involves stimulating a specific and tailored immune response against the patient’s cancer type, exposing immune cells to the entire spectrum of associated tumor antigens. However, a drawback of this tumor-specific approach lies in the necessity for sufficient tumor volume in order to obtain and prepare adequate tumor cell samples for vaccine production [5].

Alternatively, employing predetermined cell lines facilitates the creation of allogeneic cell vaccines tailored for specific cancer types.

Another strategy involves utilizing autologous dendritic cells activated through exposure to tumor-associated antigens, along with enhancing the cellular response through exposure to adjuvant molecules (see Figure 2) [6,7]. Dendritic cells, serving as antigen-presenting cells, are inoculated autologously and activated, mirroring the approach seen with Sipuleucel-T, an approved therapy for metastatic prostate cancer [8].

Protein or peptide vaccines harness tumor-associated antigens to provoke an immune response against these antigens, which exhibit heightened expression in tumor tissue compared to normal tissue types. The inclusion of these antigens alongside adjuvant immunomodulatory agents enhances the efficacy of this vaccine category [9].

Genetic vaccines employ DNA, RNA, or viral particles, wherein the genetic information is presented to antigen-presenting cells for subsequent translation into tumor-specific antigens or antigenic fragments [10]. Among these, RNA vaccines, particularly mRNA vaccines, are associated with fewer adverse effects compared to DNA vaccines due to their more rapid degradation within the target organism.

Cancer immunotherapy and immunoprophylaxis represent promising applications for synthetic RNA technology. However, challenges arise from its in vivo instability and rapid clearance within the body. To achieve the desired therapeutic effects, mRNA molecules must reach specific target cells and produce sufficient levels of the target protein to elicit a systemic immune response. The encapsulation of tumor RNA molecules within liposomal nanoparticles offers crucial protection, significantly enhancing the efficacy of RNA vaccines. The optimal size of nanoparticles may enable mRNA condensation, shield them from degradation and rapid clearance, prevent unintended delivery to non-target sites and also mediate the interaction with adjuvant molecules, which are capable of modulating and enhancing the specific immune response induced by mRNA vaccines [11].

Functionalized nanoparticles can be synthesized by coating biodegradable polymer nanoparticles with cancer cell membranes [12]. These carrier nanoparticles serve as effective antigen delivery systems, augmenting or facilitating antigen uptake by antigen-presenting cells like dendritic cells and macrophages. By incorporating multiple membrane antigens common to cancer cell membranes, these cancer cell membrane-coated nanoparticles exhibit immunogenic or surface drug delivery functionalities.

In the context of cancer immunotherapy, endogenous antigen-carrying nanoparticles (EAC-NPs) can potentiate their immunogenic effect through the inclusion of immunological adjuvants such as monophosphoryl lipid A [13]. This leads to enhanced antigen uptake for immune presentation and subsequent activation of APCs and then triggers CTL and long-term memory immune responses (Figure 3).

## 2. Delivery of Nanoparticle-Based Cancer Vaccines

Nanoparticles are promising vectors for antigen delivery in cancer vaccines due to various advantages, including prolonged biological activity, enhanced bioavailability, antigen protection from degradation and controlled antigen release. Cancer vaccines delivered via nanomaterials can be adjusted to desired immune profiles by optimizing the physicochemical properties of the nanomaterial carriers, modifying the nanomaterials with targeting molecules or co-encapsulating them with immunostimulators. NPs can function as both a delivery system and an immune-stimulating adjuvant [14]. To develop vaccines with desired immunogenicity, a comprehensive understanding of the factors that influence immune responses is necessary. Having greater insight into the correlation of NPs with the immune system would yield useful information about the possible health benefits and toxicity elicited by the NPs inside the body. The vaccine antigen is either enclosed within or attached to the surface of the NP. By encapsulating the antigenic material, NPs offer a means of delivering antigens that may otherwise degrade rapidly upon injection or induce a short-lived, localized immune response. The conjugation of antigens onto NPs enables the presentation of the immunogen to the immune system in a manner like that of the pathogen, thereby stimulating a comparable response. Furthermore, NPs not only facilitate targeted delivery of antigens but also allow for the sustained release of antigens to maximize exposure to the immune system [15]. Additionally, researchers are investigating the potential of NPs to deliver vaccines through unconventional methods such as topical, inhalation, or optical delivery, as well as the combination of multiple antigens within a single particle to provide protection against multiple diseases [16]. 

Only a small number of nucleic vaccine delivery systems based on nanomaterials have achieved success in their progression to clinical trials, and none have received approval for usage thus far. In human trials, strategies involving nucleic acid vaccines have demonstrated efficacy in inducing specific humoral and cell-mediated immune responses. However, there are limitations when considering therapeutic applications, primarily due to the significantly higher dosage requirements in humans compared to animals [17]. One example of a nanomaterial-based adjuvant that has been investigated in a clinical setting is Vaxfectin, a cationic liposome capable of ionically attaching to DNA and augmenting the immune response against H5N1 influenza-associated proteins, including HA, nucleoproteins and viroporins. 

## 3. Types of Immunostimulating Nanoparticles

Nanoparticles with immunostimulatory effects are widely used in cancer immunotherapy, as summarized in Table 1 and discussed in the following sections.

Dendrimers have shown promise in stimulating the immune system. Xu et al. used guanidinobenzoic acid (DGBA)-modified polyamidoamine dendrimers to co-deliver OVA and CpG-ODN, which were efficiently taken up by dendritic cells (DCs) and promoted their maturation and antigen presentation [18]. This nanovaccine induced CD8+ T-cell immune responses and demonstrated prophylactic efficacy against B16-OVA melanoma. In combination with immune checkpoint blockade therapy (ICBT), this vaccine showed synergistic effects on T-cell antitumor response. Chen et al. developed methoxy polyethylene glycol-decorated dendrimer-entrapped gold nanoparticles (PEG-Au DENPs) for CpG-ODN delivery to DCs, promoting DC maturation and activating T cells for an adaptive antitumor immune response [19]. 

Liposomal nanoparticles are very useful vehicles in pharmacology, but also as a means of delivering antigens to antigen-presenting cells (APCs) or with an adjuvant role in mediating the specific immune response. Liposomes modified with pH-sensitive dextran and loaded with ovalbumin were efficiently absorbed by dendritic cells, causing an effective antitumor immune response in E.G7-OVA tumor-bearing mice [20]. Also, liposomes can be used to deliver long synthetic peptides to dendritic cells, activating antigen presentation and the immune response mediated by CD8+ cytotoxic T cells [21]. 

Superparamagnetic iron oxide nanoparticles are useful in oncological diagnosis, but they are capable of therapeutic effects, being useful in cancer theranostic applications [22]. Luo L and colleagues proposed the loading of iron oxide nanocomposites with OVA in order to efficiently mature dendritic cells and activate T cells. They showed that these nanocomposites induced a competent antitumor immune response through the simultaneous activation of macrophages. Applied to murine models, these biofunctionalized nanocomposites had significant antitumor effects for B16-OVA tumors [23]. 

Micelles have shown promise as effective nanocarriers in enhancing the efficacy of cancer vaccines by delivering antigens and adjuvants. Zeng et al. used polymer hybrid micelles (PHMs) to encapsulate melanoma antigen peptides and TLR-9 agonists, observing successful lymph node targeting and payload internalization by dendritic cells [24]. This co-delivery system stimulated antigen-specific CD8+ T-cell immune responses and demonstrated potent antitumor effects in a lung metastatic melanoma model. In another study, Li et al. developed PHMs using specific polymers and observed that these micelles effectively induced stronger antigen-specific CD8+ T-cell immune responses and antitumor efficacy compared to mixtures of free antigens and adjuvant [25]. In addition, carboxylated polymer mixed micelles were used to co-deliver antigens and TLR-7 agonists, resulting in enhanced dendritic cell maturation, cytokine secretion, and antigen cross-presentation, ultimately leading to a potent antigen-specific immune response. Furthermore, immunization with these nanovaccines significantly inhibited tumor growth in experimental mice [25]. 

Silicon nanoparticles are used due to their acceptable biocompatibility and characteristic porosity in various fields such as bioimaging, tumor localization, or transport of vaccine or drug molecules [26]. Furthermore, because of their low cytotoxicity and easily adaptable morphology, gold nanoparticles are used in macrophage activation and T-cell response triggering. In 2019, Ong et al. showed that mesoporous silicon nanoparticles (MSN) decorated with gold nanoparticles can be loaded and can deliver large amounts of CpG-ODNs to the tumor. Thus, a specific antigen was generated at the tumor level, which was processed and presented by tumor-infiltrated dendritic cells, activating and triggering a specific immune response [27]. 

Liu et al. fabricated a mesoporous silicon nanoparticle (MSN) engraved with polyethyleneimine and then used it as a nucleotide delivery agent in an experimental model. Using the tyrosinase-related protein-2 molecule as an antigen, they obtained an increased cellular absorption of the antigen and dendritic cell maturation, confirmed by high levels of pro-inflammatory cytokines and the costimulatory molecules of the immune response, CD86 and CD83 [28]. Cha and colleagues prepared MSNs with a particle size of 100–200 nm, used OVA as an antigen and unmethylated CpG as a TLR9 agonist. They showed that, compared to free OVA, this nanoparticulate complex increased the expression of CD86, necessary for the priming of cytotoxic T lymphocytes, together with the major histocompatibility complex I and produced the highest level of antigen-presenting cells (APCs) [29] (Figure 4). 

Carbon nanotubes have also been used to induce immunostimulatory effects, both in vitro and in vivo. Dong et al. assembled an antigen delivery system using mannose-modified multi-walled carbon nanotubes that binds specifically to the mannose receptor on the dendritic cell membrane [30]. Loading the nanotubes thus prepared with an antigen, they showed that this system has low cytotoxicity and demonstrated the efficient incorporation of the nanotube–antigen complex at the level of dendritic cells and their maturation in order to release cytokines in vitro. Xia et al. developed a nanodelivery system for Cytosine-Guanine Oligodeoxynucleotides (CpG ODNs) based on multi-walled carbon nanotubes (MWCNTs) conjugated with H3R6 polypeptide. The in vivo anticancer efficacy study on RM-1 tumor-bearing mice demonstrated that this nanotube system could deliver immunotherapeutics to the tumor site and could suppress tumor growth [31]. 

Nanoemulsions (NEs) are used as adjuvants or antigen-delivery vehicles to enhance antitumor immune responses. Tailored NEs functionalized with the C-type lectin receptor (Clec9A) have been developed for antigen-specific immunotherapy by efficiently targeting and activating DCs and inducing antigen-specific T-cell responses [32]. Encapsulation of tumor antigens such as HPV16 E6/E7 in Clec9A nanoemulsions inhibited tumor growth and stimulated strong immune responses. NEs loaded with toll-like receptor 7/8 (TLR7/8) agonists and tumor antigens activated DCs and T cells while modulating the immunosuppressive tumor microenvironment. Combining NE treatment with immune checkpoint blockade therapy (ICBT) synergistically induced antitumor immune responses [33]. 

Nanogels have emerged as effective antigen or protein delivery systems in cancer immunotherapy. Wang et al. developed pH-sensitive galactosyl dextran-retinal (GDR) nanogels for targeted delivery of an MHC class I antigen into dendritic cells (DCs), enhancing DC maturation and antigen uptake [34]. Additionally, cationic dextran nanogels enabled its intracellular release in a reductive environment, promoting DC maturation and generating strong antitumor responses when combined with the adjuvant poly (I:C) [35]. Carboxyl group-modified cholesterol-bearing pullulan self-assembly nanogels have also been developed for antigen delivery into DCs, inducing significant adaptive immune responses [36]. 

Polymeric nanoparticles are among the most used due to their biocompatibility, biodegradability, loading capacity, stable chemical properties and water solubility. Of these, PLGA (poly(lactic-co-glycolic acid)), PGA (polyglutamic acid), PLG (poly lactide co-glycolic), PEG (polyethylene glycol), PEI (polyethyleneimine) or chitosan are among the most used as adjuvants of anticancer vaccines due to their immunostimulatory effect. In association with the toll-like receptor 7/8(TLR 7/8), PLGA nanoparticles lead to a significant increase in the expression of the TLR agonist and to a more intense stimulation of dendritic cells [37]. Subcutaneous administration leads to a concentration of nanoparticles in the lymph nodes, where they activate the immune response mediated by dendritic cells and cytotoxic CD8+ cells. In combination with TLR agonists, PLGA NPs lead to a significant increase in the antitumor immune response of anticancer vaccines in the murine experimental models of da Silva et al. [38]. 

Protein nanoparticles have demonstrated their effectiveness as vaccine platforms for delivering tumor antigens and adjuvants, thereby eliciting a potent antitumor immune response. In a study by Molino et al., biomimetic protein nanoparticles were successfully utilized to co-deliver peptide epitopes and CpG-ODN activators to dendritic cells (DCs) [39]. This approach led to heightened and sustained activation of CD8+ T cells, along with improved antigen cross-presentation. Another study revealed that the concurrent administration of melanoma-associated gp100 epitope and CpG-ODN, utilizing viral mimicking protein nanoparticles, substantially enhanced CD8+ T-cell proliferation and secretion of IFN-γ [40]. 

Virus-like particles (VLPs) have emerged as a promising nanovaccine platform for enhancing cancer immunotherapy. Lizotte et al. have shown that self-assembled VLPs derived from cowpea mosaic virus (CPMV) can significantly reduce lung melanomas and induce a potent systemic antitumor immune response in mice [41]. Also, encapsulation of CpG-ODNs into VLPs derived from cowpea chlorotic mottle virus (CCMV) has enabled targeted delivery to tumor-associated macrophages (TAMs) in the tumor microenvironment, enhancing their phagocytic activity and promoting more effective antitumor responses both in vitro and in vivo [42]. 

## 4. Vaccines in Digestive Tract Cancers 

In 2003, Liang Wei et al. published a study describing the use of an autologous vaccine using tumor cells combined with the Newcastle disease virus in patients with malignant tumors of the digestive tract who underwent surgical and radiochemotherapy treatment, obtaining an increase in the average survival period in the long term within one year [43]. 

Different types of vaccines have been tested in colorectal cancer, from peptides such as carcino-embryonic antigen (CEA), melanoma-associated antigen (MAGE) or mutant neoantigens such as mutant KRAS peptide to combinations of peptide molecules of tumor-associated antigens and amputated BCG vaccine [44]. Even if in experimental studies they had antitumor effects and determined an immune response, these vaccines did not develop a significant antitumor response in phase I and II clinical trials. 

Dai et al. used a nanoliposome–tumor RNA complex with RNA extracted from colorectal cancer tumor cells (CT-26) and developed it into a vaccine with significant antitumor immunological efficiency that was demonstrated in murine experimental models [45]. The antitumor effects of oxaliplatin in association with this RNA vaccine were also improved. 

Mohammad Ariful Islam and collaborators demonstrate a concept consisting of a nanoparticle associated with an mRNA-based vaccine using an experimental murine model with tumor allografts of prostate cancer and colorectal cancer expressing ovalbumin [46]. 

In 2013, Kimura et al. presented a clinical study conducted in patients diagnosed with advanced adenomas of the colon who were administered a vaccine containing a tumor-associated antigen, MUC1 [47]. This is a glycoprotein identified as a tumor-associated antigen. After administration, 44% of subjects developed anti-MUC1 antibodies. In the patients who did not present an adequate immune response, a peculiarity was discovered: they had a significantly higher pre-vaccination concentration of suppressive myeloid cells in the peripheral blood mononuclear population of leucocytes. This type of cell has an intense suppressive role in the immune response mediated by T lymphocytes [48]. 

Wang et al. presented an experimental model of polydopamine nanoparticles carrying tumor cell lysate as a potential vaccine for colorectal cancer immunotherapy [49]. Polydopamine nanoparticles (NP-PDAs) were prepared by self-polymerization of dopamine, on the surface of which the product obtained by tumor cell lysis (TCL) was attached. The loading capacity was 0.96 mg TCL at 2 mg NP-PDAs, and the resulting loaded nanoparticles had a size of 241.9 nm, perfect storage stability and negligible cytotoxicity against dendritic cells. Tumor-bearing mice vaccinated with tumor lysate-loaded nanoparticles showed significant delays in tumor progression due to enough TCLs and M1 tumor-associated macrophages, as well as the deficient number of immunosuppression-related cells in the tumor tissues. Moreover, hollow NP-PDAs demonstrated an ability to modulate dendritic cell maturation and delayed tumor development by facilitating the production of activated T cells and decreasing the subpopulation of myeloid-derived suppressor cells in the tumor microenvironment [50]. Nanoparticles show the ability to protect the antigen against natural degradation mechanisms inside the body until delivery to the target cells [51]. They can serve as a reservoir for the controlled release of antigen, increasing its availability for immune cells and, implicitly, the intensity and quality of the immune response. At the same time, they can modulate the type of immune responses induced when used alone or in combination with other immunostimulatory compounds [52]. 

Tumor-associated antigens (TAAs) and activated dendritic cells have been incorporated into experimental vaccines against gastric cancer, but with limited effectiveness [53]; the vaccine models studied by Ajani et al. or Fujiwara showed clinical efficacy was not observed [54,55]. 

The modern therapeutic vision in gastric cancer combines antitumor therapy with the use of nanoparticles with the role of modulating the tumor microenvironment, the degradation of the extracellular matrix and inhibition of tumor angiogenesis through the processes of lipid peroxidation, apoptosis or autophagocytosis. Preliminary studies suggest that this approach is effective in reducing the rate of resistance to cytostatic drugs and overcoming the therapeutic barriers associated with their toxicity [56]. 

CD44 cleavage, shedding and elevated levels of soluble CD44 in the serum of patients is a marker of tumor burden and metastasis in several cancers, including colon and gastric cancers. The expression of CD44 isoforms can be correlated with tumor subtypes and be a marker of cancer stem cells. Cai et al. built a model of nanoparticles based on a metal-organic framework with the aim of combining photodynamic therapy, antihypoxic signaling and a CpG-type adjuvant to obtain an in situ antitumor vaccine. These NPs, self-assembled from meso-Tetra(4-carboxyphenyl)porphine (H2TCPP) ions and zirconia with hypoxia-inducible factor signaling inhibitor and loaded with immunological adjuvant (CpG ODNs) and hyaluronic acid coating on the surface, specifically target cancer cells overexpressing the CD44 receptor. Photodynamic therapy generates multiple tumor antigens at the tumor level through cell destruction without determining the hypoxic signaling effects suppressed by the presence of signaling inhibitor, and the presence of the CpG ODN adjuvants generates a strong antitumor immune response, which eliminates residual tumor cells [57,58]. 

In the case of hepatocellular cancer, TAA-based vaccines have not been proven to have clear clinical benefits. Alpha-fetoprotein (AFP) is expressed in 80% of hepatocarcinomas, so Butterfield et al. included the AFP vaccine in phase I and II trials. Even if it determined the appearance of an immune reaction of T cells, the antitumor effect of the vaccine could not be demonstrated [59]. Peptide vaccines have been tried in combination with chemotherapy in limited groups of patients, with inconsistent results from the perspective of the correlation of the immune response with the clinical antitumor response. 

Linlinh He et al. presented a novel HCV vaccine strategy that combines E2 glycoprotein optimization and nanoparticle display to stimulate a robust B-cell response to vaccination. These new E2 cores not only retained the native-like structure but also showed improved thermostability and antigenicity. Displayed on nanoparticles of various sizes, they were used as carriers for HIV-1 gp140 trimers. These E2-based nanoparticles demonstrated high yield, high purity, and improved antigenicity. In mice immunized with the novel E2 core and nanoparticle constructs, they showed a superior immunogenicity of E2p-based vaccine constructs [60]. Optimizing the composition and physicochemical characteristics of the nanoparticles may allow safe delivery to the liver tumor [61]. 

CA 19-9 is a tumor-associated antigen intensively expressed in pancreatic cancer; therefore, it was used as a target antigen to produce anti-CA 19-9 antibodies, and these, in turn, represented a protective factor against the progression of pancreatic cancer within murine experimental models. That is why this antigen can be a candidate for the development of an antitumor vaccine against pancreatic cancer [62]. 

Antigens specific to pancreatic cancer tumor cells show an important genetic variability due to their genetic instability. These antigens can be used to sensitize dendritic cells, which subsequently cause an immune response from CD4+ lymphocytes [63]. MUC-1 peptide is also able to determine an antitumor immune response mediated by dendritic cells in patients with metastatic pancreatic cancer [64]. 

## 5. Functionalized Nanoparticles in Digestive Cancer Vaccines

A broad range of NP delivery systems as vaccine carriers or vaccine adjuvants have been studied, and each presents benefits over existing approaches of vaccine delivery, as NPs can easily encapsulate target antigens, proteins, peptides, or nucleic acids and provide sustained release or target-specific release of the vaccine payload into immune cells after crossing biological barriers and long-lasting immunological effects. Patients with GI cancers have antigen-specific tolerance to cancer. Immune tolerance involves various T lymphocytes that are either immunogenic or tolerogenic. Therapeutic vaccines increase both types of T lymphocytes and, thus, might amplify cells that are involved in both tumor tolerance and rejection, which nullifies the therapeutic efficacy [65]. 

In the prophylactic setting, such as the HBV vaccine, these immune responses help confer protection; however, in the therapeutic setting, the vaccine-induced immune response fails to be clinically beneficial due to the tumor microenvironment comprising tolerogenic lymphocytes that have either infiltrated or are in the vicinity of the tumor [66]. 

There exist numerous immune processes of varied antitumor leukocytes, and tumor cells employ various strategies to evade the immune response. The tumor microenvironment plays a role in determining the activated immunosuppressive pathways that suppress antitumor immunity [67]. These pathways involve immune checkpoint receptors on effector T cells and myeloid cells, as well as the release of inhibitory cytokines and metabolites. Therapeutic approaches that target these pathways, particularly immune checkpoint receptors, have the potential to induce durable antitumor responses in patients with advanced cancer [68]. Incorporation of immune modulators exhibits significantly greater efficacy in terms of tumor growth regression and prevention of metastasis in murine models of breast and colorectal cancers [69]. 

Advanced cancer immunotherapy involves the systemic administration of drugs. One strategy for improving antitumor efficiency is to identify the potential for the transport and delivery of compounds with immunological functions directly at the level of the lymph nodes. Thus, interest in the role of biofunctionalized nanoparticles in cancer immunotherapy has increased. Nanostructured lipid carriers are a pharmaceutical vector with significant pharmacodynamic advantages and reduced toxicity, and they offer increased bioavailability [70]. As these nanoparticles are largely captured in macrophages and dendritic cells, they specifically target the lymph nodes, and the primary antitumor immune response of these cells is activated [71]. 

For a generalized clinical approach in individual cancer therapy, it is necessary to find specific tumor peptides adapted individually to the patient. This peptide complex can be obtained by lysing tumor cells. Following this design, nanoparticles can be functionalized using an entire peptide complex derived from primary tumor cell lysis [72,73]. 

A group led by Carolin Hesse from Essen, Germany, conducted a study using the murine experimental model, in which a Ca phosphate nanoparticle (CaP) was biofunctionalized with tumor antigens from tumor cell lysate and CpG adjuvant molecules. These functionalized NPs showed in vivo a strong suppressive effect on tumor cell mass growth by activating immunity mediated by specific antitumor CD8+ T cells [74]. Using the murine model with a xenograft tumor expressing the viral antigen hemagglutinin (HA), it was shown that the administration of CaP nanoparticles functionalized with this peptide (HA) and the adjuvant molecule CpG was highly effective in enhancing the antitumor T-cell response and suppressing the progression of the tumors. Mice were subcutaneously transplanted with CT26 tumor cells and therapeutically vaccinated with CaP nanoparticles containing CpG and HA peptides or a whole tumor peptide from a cell lysate. Therapeutic vaccination of tumor-bearing mice with CpG and CaP nanoparticle-delivered tumor lysate significantly suppressed tumor growth. Vaccination with soluble and lysed CpG also had a statistically significant effect (*p* < 0.01), although it induced only a 1.9-fold decrease in tumor volume compared to the 3.2-fold decrease after vaccination with CaP nanoparticles. In contrast, immunization of CT26 tumor-bearing mice with nanoparticles functionalized with CpG and HA or with soluble CpG and HA did not significantly alter tumor growth, underscoring the need for tumor-associated antigens in the vaccine [74]. 

Long Chen et al. developed a nanoparticulate delivery system for antigen and an adjuvant based on the cytoplasmic membrane of *E. coli* and the membranes of tumor cells, with the aim of activating a sufficiently robust specific antitumor response without notable side effects. The introduction of this *E. coli* membrane adjuvant in the nanoparticle formula of the antitumor vaccine succeeded in increasing immunogenicity through the maturation of dendritic cells and the activation of T lymphocytes from the splenic level. This hybrid vaccine based on a nanoparticle loaded with antigens from the tumor cell membrane and with the immunogenic adjuvant of the cytoplasmic membrane of *E. coli* proved its effectiveness in the murine experimental model with CT26 colon tumor and 4T1 breast tumor. It also caused a prolonged specific antitumor immune response in CT26 tumors, mediated especially by CD8+ T cells and NK cells [75]. 

Nanoparticles of poly(lactic-co-glycolic acid)—a polymer known for its ability to protect the antigen encapsulated in it from enzymatic proteolytic action—used as delivery vehicles of the antigen from the tumor lysate to the target dendritic cells were adapted to an experimental murine model of gastric cancer immunotherapy [76]. An important increase in antitumor immunological stimulation mediated by dendritic cells was demonstrated by the presentation of tumor antigen encapsulated in polymeric nanoparticles. 

Cholesterol introduced into the hydrophobic polysaccharide pullulan (ChP) forms spontaneously aggregated nanoparticles that can act as a vector for a vaccine based on tumor antigens. This system leads to the activation of CD4+ and CD8+ T lymphocytes. The NY-ESO-1 antigen (New York esophageal squamous cell carcinoma 1) is expressed exclusively in testicular, placental or tumor tissue, which makes it an ideal candidate for use in cancer immunotherapy [64]. Likewise, the HER2 protein has been used repeatedly in anticancer vaccines without causing any significant side effects. Both antigens led to the activation of antigen-specific cellular immunity mediated by CD4+ and CD8+ T lymphocytes [77]. 

ChP-NY-ESO-1 complexes proved ineffective as an anticancer vaccine in clinical trials due to interactions with the immunoinhibitory tumor microenvironment. The addition of molecules with an adjuvant role, such as TLR stimulants (OK-432, CpG or poly-ICLC), can make this vaccine more effective in triggering an adequate immune response without increasing the rate of side effects. 

Ishihara et al. conducted a phase 1 clinical study in which they used the ChP-NY-ESO-1 protein vaccine and an adjuvant derived from Propionibacterium Acnes, MIS416, with a role in activating the immune response mediated by toll-like receptors 9 and NOD 2 [78]. Although the safety and tolerability profiles were satisfactory, no complete or partial response to this immunotherapy was observed. In general, tumor antigens used only in combination with molecules with an adjuvant role did not demonstrate a satisfactory antitumor effect. The same results were obtained using another adjuvant, polyinosinic-polycytidylic acid (poly I/C), mixed with the stabilizers carboxymethylcellulose and polylysine (Poly ICLC), which is a ligand for the toll-3 receptor [79]. In the phase 1 clinical study in patients with advanced and recurrent esophageal cancer, the side effects were relatively minor, but no significant tumor response was demonstrated in the patients in the two groups, study and control. However, both groups, both those vaccinated with ChP-NY-ESO-1 alone and those with the combination with Poly ICLC, developed comparable antibody titers, but higher in the group with the combined vaccine. In the murine model, the addition of anti-PD-1 monoclonal antibodies, Nivolumab, in the nanoparticulate vaccine combination led to the suppression of tumor growth in tumors expressing the NY-ESO-1 antigen. The association of an immune checkpoint inhibitor could lead to significantly improved antitumor results in clinical trial scenarios [80]. 

There exist numerous immune processes of varied antitumor leukocytes, and tumor cells employ various strategies to evade the immune response. The tumor microenvironment plays a role in determining the activated immunosuppressive pathways that suppress antitumor immunity [68]. These pathways involve immune checkpoint receptors on effector T cells and myeloid cells, as well as the release of inhibitory cytokines and metabolites [69,81]. 

Xia Dong et al. have developed a nanovaccine for cancer immunotherapy that is composed of self-crosslinked nanoparticles serving as antigens and transport as natural carriers of CpG ODNs, thereby allowing for continuous immune response stimulation [72,82]. As a result, a robust immune response was achieved in both in vitro and in vivo settings, including dendritic cell maturation, T-cell activation, and production of IFN-gamma. 

## 6. Conclusions

Cancer vaccines must be safe, effective and affordable. We have seen many examples of vaccines that are effective in developing an antitumor immune response. The unique properties of nanoparticles in the development of these vaccines are substantial due to their safety, controlled release, targeting of DCs and improved antigen uptake, as well as enhanced immunogenicity. However, most tumors that become clinically evident have developed a tumor microenvironment that protects them from autoimmunity and inflammation, so previous failures of cancer vaccines are primarily due to the inability of the generated immune response to reach its full potential, with the immunoregulatory mechanisms developed in the tumor tissue being extremely effective in counteracting the immune system. Therefore, the combination of cancer vaccines, together with treatments that restrict the cancer microenvironment interaction, would further improve the immunotherapeutic outcome and have been extensively evaluated both preclinically and clinically [83]. 

One of the newest and most efficient methods is the blocking of checkpoint inhibitors, which leads to the amplification of the immune response mediated by T cells; the latter are considered the main effectors of cancer vaccines [84]. 

Several research in the literature and clinical trials have suggested that more than 70% of cancers (especially solid tumors) are poorly infiltrated by CD8+ T cells, which are the main causes of therapeutic failure, not only for cancer vaccines but also for point blockade control and CAR-T cell therapy [85]. Thus, modulators of the tumor environment, such as small kinase inhibitors, antibodies or RNAi, that modulate the suppressive immune environment (i.e., siRNA against STAT3, small molecules against CXCR4, tyrosine kinase inhibitor against vascular endothelial growth factor) are very useful additions to the cancer vaccine complex [14]. 

Combining the vaccine with chemotherapeutic agents is another option for improving antitumor efficiency. Their immunomodulatory properties can improve the vaccine-mediated antitumor immune response. Adapted to specific chemotherapeutic treatment schemes, nanoparticles, through their versatility, can help combined therapy reach its full potential. 

NPs are good candidates for the delivery of cancer vaccines due to their safety and versatility. The design considerations discussed in this review provide guidance for improving cancer vaccine potency. However, due to the extensive suppressive immune microenvironment, cancer vaccines have difficulties preventing disease recurrence, which requires further regulation of the suppressive tumor microenvironment to enhance in situ T-cell penetration and activation. Therefore, we envision that combination therapy, together with intelligent NP design, can overcome many of these obstacles. 

In conclusion, we consider that nanoparticles, due to their versatility and safety, are very suitable candidates as platforms for the design of vaccines against cancer. However, the interaction with the tumor microenvironment can have an immunosuppressive effect, and the solitary effectiveness of vaccines decreases significantly. This fact expresses the need for a favorable modulation of the immunosuppressive tumor microenvironment to improve the penetration and activation of T lymphocytes in situ. Combined therapy and the complex design of biofunctionalized nanoparticles by associating immunomodulatory molecules should be the way forward in the development of future vaccines against cancer. 

## Figures and Tables

**Figure 1 pharmaceutics-16-00410-f001:**
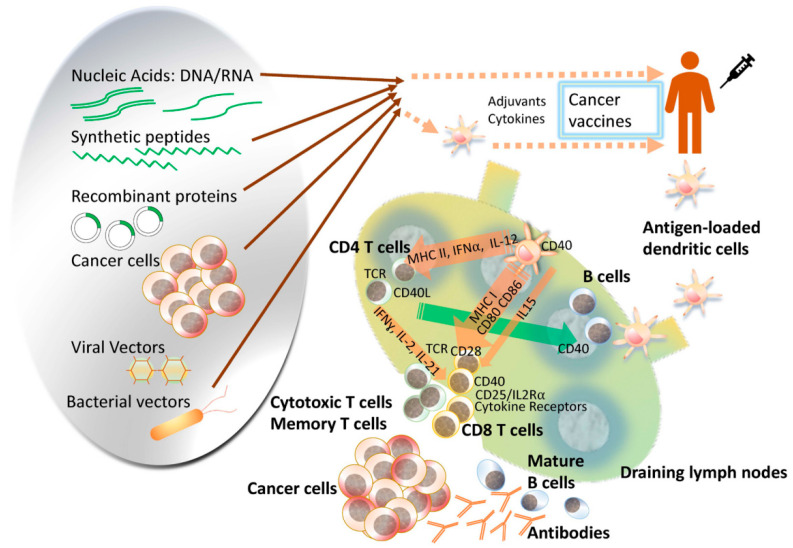
Cancer vaccines can be composed of various platforms to deliver specific tumor antigens. These platforms offer advantages such as simpler manufacturing and flexibility in vaccine delivery. Cell-based vaccines, like dendritic cell (DC) vaccines, allow targeted loading of antigens and manipulation in vivo. However, standardizing manufacturing and quality assessment poses challenges. As our understanding of the immune system grows, there is potential for more efficient and intelligent design of cancer vaccine platforms. These vaccines can be used alone or in combination with other cancer therapies, expanding the scope of cancer immunotherapy as the fourth pillar in oncology, alongside surgery, chemotherapy, and radiation. Reproduced with permission from [4].

**Figure 2 pharmaceutics-16-00410-f002:**
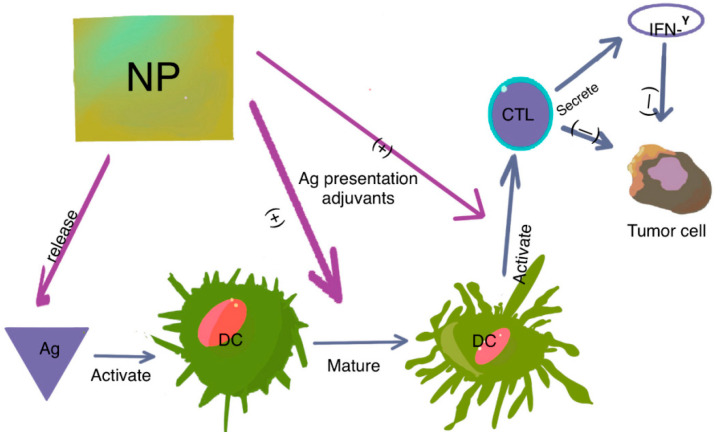
The role of DCs in tumor immunity. DCs present antigens to naïve T cells, leading to T-cell activation and transformation into cytotoxic T lymphocytes (CTLs). CTLs then attack tumor cells through direct killing or IFN-γ-dependent pathways. Nanoparticles (NPs) modified with antigens and adjuvants have two main functions: they specifically deliver antigens to DCs, and they promote DC maturation and CTL activation, either by antigen presentation or with the help of adjuvants. This results in the activation and expansion of CD4+ and CD8+ T cells, granting them cytotoxic abilities or helper functions, such as IFN-γ secretion.

**Figure 3 pharmaceutics-16-00410-f003:**
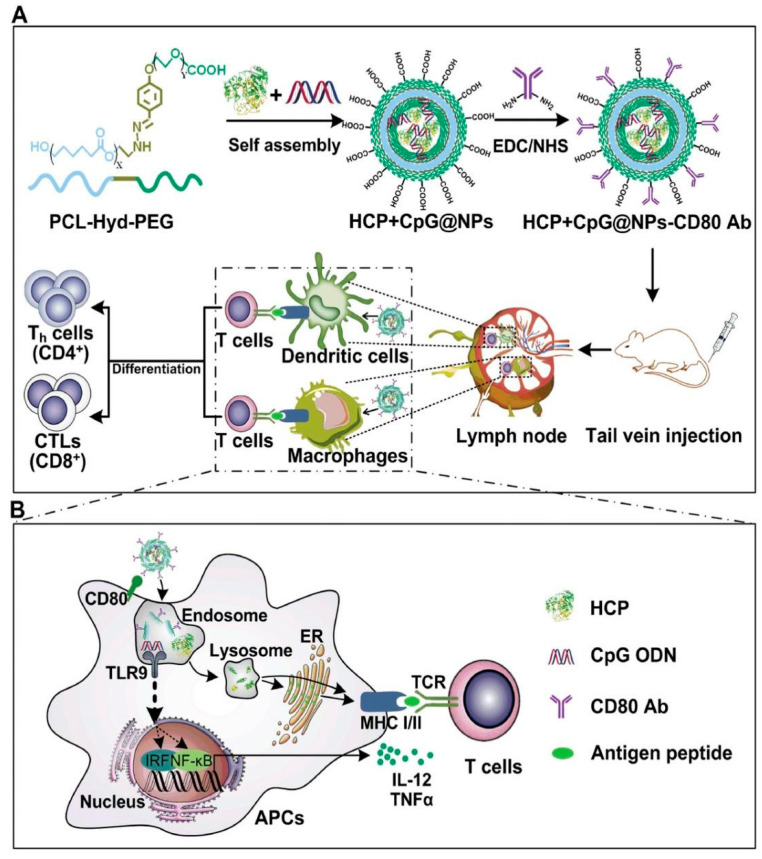
Schematic illustration of the endogenous antigen-carrying nanoparticles (EAC-NPs), EAC-NP formation and the mechanisms of EAC-NP-induced cancer immunotherapy. (**A**) Schematic illustration of the preparation of HSP70-chaperoned polypeptides HCP+CpG@NPs-CD80 Ab vesicles and induction of T-cell immune responses. (**B**) Partial magnification of (**A**). Once phagocytosis occurs, antigen-presenting cells are activated through two signaling pathways: antigen signaling and TLR signaling. After activation, APCs deliver antigen signaling to T lymphocytes, which differentiate into helper T (Th) cells and CTLs and even produce memory T cells. Reproduced with permission from [13].

**Figure 4 pharmaceutics-16-00410-f004:**
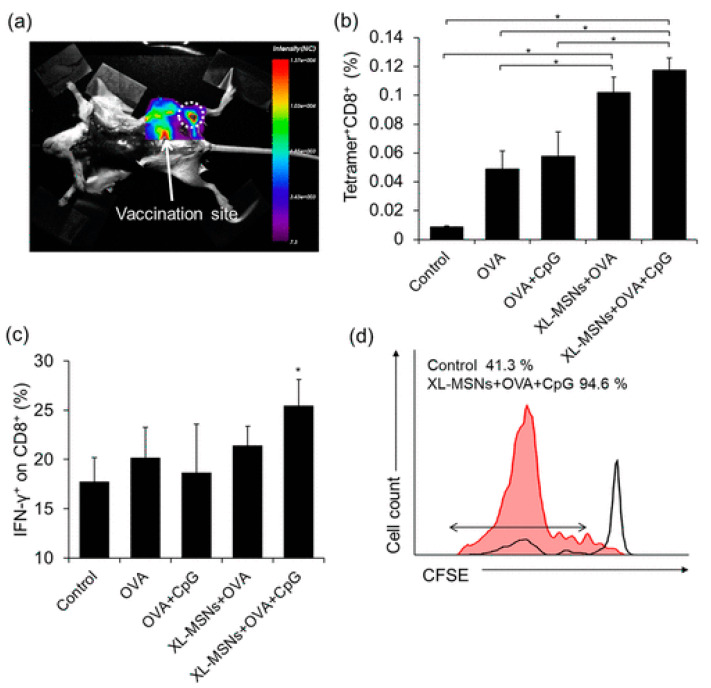
(**a**) Fluorescent images of mouse injected with RITC-labeled XL-MSNs subcutaneously on abdomen region, showing targeting of XL-MSNs to the draining lymph node (white dotted circle). (**b**) OVA-specific and (**c**) intracellular cytokine-secreting CTLs in the spleens of vaccinated mice measured in flow cytometry (*n* = 6). Error bars, mean ± s.d. * *p* < 0.05. (**d**) Proliferation of CFSE-labeled OVA-specific CD8+ T cells in the lymph node (red line: XL-MSN + OVA + CpG, black line: control), reproduced with permission from [29].

**Table 1 pharmaceutics-16-00410-t001:** Nanoparticles with immunostimulatory effects in cancer immunotherapy.

Type of NP	Associations	Effects
Dendrimers	OVA CpG-ODNs	Induce a higher T cell-mediated immune response [18,19]
Liposomes	OVA CpG-ODN SLPs antigens	Increases antigen-specific immunity mediated by DCs and CD8+ T cells [20,21]
Magnetic/Iron oxide NPs	OVA IFN-γ poly (I:C) imiquimod	Accumulation of NPs at the tumor site, which stimulates antitumor immune response [22,23]
Micelles	Trp2 CpG-ODN OVA	Antigen-specific humoral and cellular immune response [24,25]
MSNs	Doxorubicin (DOX) OVA CpG-ODN	Induces both antibody and cell-mediated immune responses, strong CD8+ T-cell response and enhanced antitumor activity [26,27,28,29]
MWCNTs	OVA CpG-ODN NY-ESO	Strong CD4+ T, CD8+ T cell-mediated immune response [30,31]
Nanoemulsions	TLR7/8 agonists OVA long peptide of E7 antigen	Enhances the efficacy of cancer immunotherapy by activating DCs and T cells and reprogramming TME [32,33]
Nanogels	OVA and poly (I:C)	Effective delivery of antigen to DCs with strong antigen-specific adaptive immunity [34,35,36]
PLGA	TLR 7/8 agonist Poly (I:C)	Enhances antigen-specific response by increased uptake of NPs by DCs [37,38]
Protein NPs	NY-ESO-1 MAGE A3 CpG-ODN	Significant antigen-specific cell-mediated immune response [39,40]
VLPs	CpG-ODNs	Enhances the efficacy of CpG-ODNs against tumors and induces a potent antitumor response [41,42]

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
