# Peer review of "Role of Biofunctionalized Nanoparticles in Digestive Cancer Vaccine Development"

_pharmaceutics, 2024, doi:10.3390/pharmaceutics16030410_

Round 1
Reviewer 1 Report (New Reviewer)
Comments and Suggestions for Authors
The manuscript title "Role of biofunctionalized nanoparticles in the digestive cancer vaccine development" in my opinion is totally misleading. The cancers from digestive tract (colorectal, gastric, liver, oesophageal) are mentioned only after a very long description of several types of nanoparticles: their composition and what was accomplished by using them, and many of these nanoparticles were used with the model antigen OVA (no digestive cancer).
Follow a long roll-call of types of cancer vaccines (not only nanoparticles) and finally there is a description of vaccines in digestive tract: not only biofunzionalized nanoparticles but peptides, activated DCs, TAA...
The tumor microenvironment section describes (among several other informations) the co-delivery of antigens and immune checkpoint inhibitors, but its description is confusing, while many reported experiments refer to OVA model antigen.
In my opinion this manuscript put together several different approaches to cure cancer, either in pre-clinical model or in clinical trial, with well documented references but I cannot envisage any logical thread in the description of these selected papers. I invite the authors to reorganize the content of the manuscript, focusing only on a selected section of the topics described here. Moreover all the figures derive from papers mentioned and summarized in the manuscript, but not for all the figures the permission is reported.
Finally the statement reported in lanes 53-56 should be documented with references because the fact that the quality of vaccinations varies from batch to batch is a very strong assertion, and about the adverse effects a mention of reference is missing.
Author Response
The authors would like to thank the reviewers for their valuable comments. All the requested changes were incorporated in the submitted manuscript
Please receive bellow point by point comments
Reviewer 1:
Comment
The manuscript title "Role of biofunctionalized nanoparticles in the digestive cancer vaccine development" in my opinion is totally misleading. The cancers from digestive tract (colorectal, gastric, liver, oesophageal) are mentioned only after a very long description of several types of nanoparticles: their composition and what was accomplished by using them, and many of these nanoparticles were used with the model antigen OVA (no digestive cancer).
Response:
We thank the reviewer for the remark, as result an we reduced the part reffering to the types of nanoparticles and also the refferences to OVA.
Comment
Follow a long roll-call of types of cancer vaccines (not only nanoparticles) and finally there is a description of vaccines in digestive tract: not only biofunzionalized nanoparticles but peptides, activated DCs, TAA…
Response
We thank the reviewer for the remark. We reduced the general references to cancer vaccine, leaving only an introductory figure for cancer vaccines in general.
Comment
The tumor microenvironment section describes (among several other informations) the co-delivery of antigens and immune checkpoint inhibitors, but its description is confusing, while many reported experiments refer to OVA model antigen
Response:
We renounced to have a separate section about the tumor microenvironment and the relevant information was introduced in the text to emphasize the importance of complex development of nanoparticle vaccine delivery systems.
Comment
In my opinion this manuscript put together several different approaches to cure cancer, either in pre-clinical model or in clinical trial, with well documented references but I cannot envisage any logical thread in the description of these selected papers. I invite the authors to reorganize the content of the manuscript, focusing only on a selected section of the topics described here. Moreover all the figures derive from papers mentioned and summarized in the manuscript, but not for all the figures the permission is reported.
Response
We have reorganized the content of this manuscript, focusing on describing the biofunctionalization of the different types of nanoparticles (chapters 2 and 3) and how they have been used in different studies on digestive tract cancers (chapters 4 and 5).
We have obtained permisions for the figures 1,3 and 4 in this manuscript. We will send the permisions to the editor along with this revision of the manuscript. Figure 2 is a graphic representation by the authors for activating dendritic cells and their immunological effect, through functionalized nanoparticles.
Comment
Finally the statement reported in lanes 53-56 should be documented with references because the fact that the quality of vaccinations varies from batch to batch is a very strong assertion, and about the adverse effects a mention of reference is missing.
Response
We thank the reviewer for this observation. We have removed this statement, as it was not in support of the idea behind this review

Reviewer 2 Report (New Reviewer)
Comments and Suggestions for Authors
The review on the 'role of biofunctionalized nanoparticle is digestive cancer vaccines development' gives a quick and well overview on the state of the art. The literature is sufficient and necessary. The publication of the review is explicitly recommended.
1) Unfortunately there is a mistake in the reference numbers in trable1, block 2-4, and in the corresponding text, page4 block of line 83-120: the reference numbers 8-15 are wrongly used, e.g. from an earlier manuscript version. E.g. the references 8,9 in the table do not fit to liposomes, but silica, and the references of magnetic particles are 12,13. The correction needs a minor revision by the authors.
2) the fact, that mRNA induces the formation of tumor-specific material, rather then its physical input should be noted by a sentence.
3) the terms 'in vivo' and 'in vitro' are better recognized, if presented in italics.
Author Response
Reviewer 2:
Comment
The review on the 'role of biofunctionalized nanoparticle is digestive cancer vaccines development' gives a quick and well overview on the state of the art. The literature is sufficient and necessary. The publication of the review is explicitly recommended.
Response:
We thank the review for this comment and kind recomandation
Comment
1) Unfortunately there is a mistake in the reference numbers in trable1, block 2-4, and in the corresponding text, page4 block of line 83-120: the reference numbers 8-15 are wrongly used, e.g. from an earlier manuscript version. E.g. the references 8,9 in the table do not fit to liposomes, but silica, and the references of magnetic particles are 12,13. The correction needs a minor revision by the authors.
Response:
We thank the review for this observation. The references have been corrected and checked for the entire manuscript.
Comment
- the fact, that mRNA induces the formation of tumor-specific material, rather then its physical input should be noted by a sentence.
Response:
We have explained in the text that by introducing a piece of mRNA that corresponds to a tumor protein, usually a small piece of a protein found on the tumor cell outer membrane, cells produce this protein and the immune system recognizes that the protein is foreign and actrivates dendritic cells with tumor RNA, generating a systemic immune antitumor response.
Comment
- the terms 'in vivo' and 'in vitro' are better recognized, if presented in italics.
Response:
We thank the reviewer for this recommendation. We have put the terms in vivo and in vitro in italics in the text

Round 2
Reviewer 1 Report (New Reviewer)
Comments and Suggestions for Authors
The amended manuscript is more focused but some parts are approximate: see lanes 98-102: "mRNA vaccines work by introducing a piece of mRNA that corresponds to a tumor protein, usually a small piece of a protein found on the tumor cell outer membrane. By using this mRNA, cells can produce this protein. As part of a normal immune response, the immune system recognizes that the protein is foreign and actrivates dendritic cells
with tumor RNA, generating a systemic immune antitumor response."
or lane 139-142 "Compared to conventional vaccines, cancer vaccines delivered via nanomaterials can be adjusted to desired immune profiles by optimizing the physicochemical properties of the nanomaterial carriers, modifying the nanomaterials with targeting molecules, or co-
encapsulating them with immunostimulators." What is the difference between conventional and cancer vaccines? Therapeutic versus prophylactic?
Lanes 824-828 are repeated in lanes 828-832
Moreover the word "certain" referred to nanoparticle composition is ambigous
Author Response
The authors would like to thank the reviewer for their valuable comments. All the requested changes were incorporated in the submitted manuscript

Round 3
Reviewer 1 Report (New Reviewer)
Comments and Suggestions for Authors
I suggest to accept for publication the revised version of the manuscript
This manuscript is a resubmission of an earlier submission. The following is a list of the peer review reports and author responses from that submission.
Round 1
Reviewer 1 Report
Comments and Suggestions for Authors
In this article, Zdrehus et al. systematically review the types of immunostimulating nanoparticles and vaccines against cancer. They also introduce the development of vaccines in digestive tract cancers and the influence of tumor microenvironment on cancer treatment. Through this review, the authors concluded that nanoparticles are good candidates for the delivery of cancer vaccines due to their safety and versatility. In addition, combined therapy and the complex design of biofunctionalized nanoparticles by associating immunomodulatory molecules should be the way forward in the development of future vaccines against cancer due to the extensive suppressive immune microenvironment.
There are some suggestions that need to be addressed before it is considered for publication.
1. This article titled with “On the role of biofunctionalized nanoparticles in digestive cancer vaccine development”. In part “Vaccines in digestive tract cancers”, the authors introduce the development of vaccines against different types of digestive tract cancers. However, there is no sufficient information about how these vaccines were delivered, what kind of biofunctionalized nanoparticles have been used to treat digestive tract cancers, and how these biofunctionalized nanoparticles improve the efficacy of vaccines. I think the authors should make the content is more consistent with your title.
2. In line 345,Sorafenib is a small molecule, I don’t understand why the authors cite Sorafenib in this paper.
Author Response
We thank the reviewer for the kind remark, as result an entire section entitled: Delivery of nanoparticle-based cancer vaccines has been added.
Comment
In line 345,Sorafenib is a small molecule, I don’t understand why the authors cite Sorafenib in this paper.
Response:
We kindly thank the reviewer for the advice the sentence has been removed.
Reviewer 2 Report
Comments and Suggestions for Authors
Fig. 1 and especially Fig. 2 have very poor resolution.
What does it mean “Memora T cells” in Fig. 1? Is it “Memory T cells”?
Author Response
The figures resolution was adjusted according with the reviewer suggestions also the typos error was corrected.
Reviewer 3 Report
Comments and Suggestions for Authors
- this review is too preliminary, and more contents and figures/tables are required, substantial revise is mandatory.
- the structure of this review is a mess, please make it more clear, especially compare chapter 5. The tumor microenvironment with other chapters.
- what are the authors' perspectives?
- how about the delivery mean of cancer vaccine?
- There is the latest review (DOI: 10.58567/ci02010006) regarding the Nanomaterials in Tumour Therapy for author reference as a template.
- my most concern is to suggest providing more tables/figures (at least 3 comprehensive figures and 1-2 tables in a good review) to enrich this review. Additionally, please straighten the structure of this review.
- I suggest the authors reference more studies from big guys or high-profile journals.
Author Response
Comment
this review is too preliminary, and more contents and figures/tables are required, substantial revise is mandatory.
- the structure of this review is a mess, please make it more clear, especially compare chapter 5. The tumor microenvironment with other chapters.
Response:
We would like to thank the reviewer for these valuable comments the chapter 5 was completely re-written and two more figures were added to increase its importance
Comment
- what are the authors' perspectives?
- how about the delivery mean of cancer vaccine?
Response:
We thank the reviewer for this constructive suggestion a new section with the delivery and perspectives of nanoparticles as anticancer vaccines was added.
Comment
- There is the latest review (DOI: 10.58567/ci02010006) regarding the Nanomaterials in Tumour Therapy for author reference as a template.
Response:
We thank the reviewer for the provided help the paper has been added.
Comment
- my most concern is to suggest providing more tables/figures (at least 3 comprehensive figures and 1-2 tables in a good review) to enrich this review. Additionally, please straighten the structure of this review.
Response:
Two more figures were added
Comment
- I suggest the authors reference more studies from big guys or high-profile journals.
Response: The references from strong journals such as: Nature Reviews Methods Primers ; Nature Reviews Materials; Nature Biomedical Engineering ….were added
Reviewer 4 Report
Comments and Suggestions for Authors
This is good review.
The use of nanoparticles in vaccine development is discussed in report. The topic is very important to the field because nanoparticles allow excellent delivery for vaccine development. Since COVID vaccines, this is developed in many applications. It discussed combination of LNP with other immune players. The authors could add clinical trials and more clinical studies for this field .
Author Response
Comment
This is good review.
The use of nanoparticles in vaccine development is discussed in report. The topic is very important to the field because nanoparticles allow excellent delivery for vaccine development. Since COVID vaccines, this is developed in many applications. It discussed combination of LNP with other immune players.
The authors could add clinical trials and more clinical studies for this field .
Response: We deeply thank you the reviewer for the nice words. Also the suggestion is extremely welcomed so we added at the end of the paper a section that refers to the current trials.